# A specific E3 ligase/deubiquitinase pair modulates TBP protein levels during muscle differentiation

Li Li[1,2], Silvia Sanchez Martinez[1], Wenxin Hu[1], Zhe Liu[3], Robert Tjian[4,2]*

[1]Janelia Research Campus, Howard Hughes Medical Institute, Ashburn, United States; [2]Li Ka Shing Center for Biomedical and Health Sciences, California Institute for Regenerative Medicine Center of Excellence, Department of Molecular and Cell Biology, University of California, Berkeley, Berkeley, United States; [3]Howard Hughes Medical Institute, Janelia Research Campus, Ashburn, United States; [4]Transcription Imaging Consortium, Janelia Research Campus, Howard Hughes Medical Institute, Ashburn, United States

**Abstract** TFIID—a complex of TATA-binding protein (TBP) and TBP-associated factors (TAFs)—is a central component of the Pol II promoter recognition apparatus. Recent studies have revealed significant downregulation of TFIID subunits in terminally differentiated myocytes, hepatocytes and adipocytes. Here, we report that TBP protein levels are tightly regulated by the ubiquitin-proteasome system. Using an in vitro ubiquitination assay coupled with biochemical fractionation, we identified Huwe1 as an E3 ligase targeting TBP for K48-linked ubiquitination and proteasome-mediated degradation. Upregulation of Huwe1 expression during myogenesis induces TBP degradation and myotube differentiation. We found that Huwe1 activity on TBP is antagonized by the deubiquitinase USP10, which protects TBP from degradation. Thus, modulating the levels of both Huwe1 and USP10 appears to fine-tune the requisite degradation of TBP during myogenesis. Together, our study unmasks a previously unknown interplay between an E3 ligase and a deubiquitinating enzyme regulating TBP levels during cellular differentiation.

*For correspondence:
jmlim@berkeley.edu

Competing interests:
See page 17

## Introduction

The TATA-box binding protein (TBP) is one of the central players in eukaryotic transcription. TBP serves as a key subunit to facilitate transcription initiation by all three RNA polymerases in eukaryotic cells. As a part of the SL1 complex, TBP plays a role in recognizing Pol I-transcribed promoters (*Comai et al., 1994*). Similarly, TBP and its associated factors (TAFs) that make up the TFIID complex are specific for Pol II-mediated mRNA transcription (*Dynlacht et al., 1991*). Likewise, the TBP/B/BRF complex TFIIIB drives the transcription of small nuclear RNAs by Pol III (*Taggart et al., 1992*). Given this pivotal role in transcription, TBP-mediated transcriptional regulation has been extensively studied by biochemical and genetic approaches in past decades(*Hernandez, 1993*). However, only limited studies have interrogated the regulation of TBP during cell-type specification. The main reason is that cell-type specific transcription programs have generally been considered to be dictated by classic sequence-specific transcription factors (*Farnham, 2009*), while the components of the core promoter recognition machinery such as TBP were thought to be largely invariant across different cell types (*Thomas and Chiang, 2006*).

Recently, studies demonstrated that during a number of terminal differentiation processes, TBP protein levels become dramatically downregulated. Specifically, in terminally differentiated myocytes, hepatocytes and adipocytes, normally high concentrations of TBP protein as well as the canonical

**eLife digest** Most animal cells specialize to perform particular roles that contribute to the survival of the animal in different ways. For example, the cells that form our muscles are able to contract, while other cells in the body are efficient at storing fat. The different types of cells develop from unspecialized cells, but it is not clear what controls this process to form a particular type of cell in the right place at the right time.

The TATA-box binding protein (TBP) is one of a group of proteins that helps to activate the expression of genes in animal cells. Recent studies have revealed that TBP is deliberately destroyed by a group of proteins called the proteasome in muscle cells, in a type of liver cell, and in fat cells. Here, Li et al. used biochemical techniques to study the regulation of TBP during the formation of muscle cells from less specialist mouse cells called myoblasts.

The experiments show that an enzyme called Huwe1 selectively adds a tag to TBP that marks TBP for destruction by the proteasome. Another protein called USP10 acts to remove the tags to prevent TBP from being destroyed. Therefore, it appears that changes in the levels of Huwe1 and USP10 fine-tune the amount of TBP that is degraded during the formation of muscle cells.

Li et al.'s findings suggest that other proteins that are also involved in activating gene expression may also be destroyed as muscle cells form. The next step is to understand how important the degradation of these proteins is to the formation of other types of specialist cells.

components of the TFIID complex become severely reduced while other core pre-initiation complex components such as RNA Pol II remain largely unaffected (*Deato and Tjian, 2007*; *D'Alessio et al., 2011*; *Zhou et al., 2013*). In conjunction with TBP downregulation, various cell-type specific 'orphan' TAFs are switched on to help direct developmental gene expression and terminal differentiation program (*D'Alessio et al., 2009*; *Goodrich and Tjian, 2010*). Interestingly, it was found that reductions in TBP protein levels are much more pronounced than decreases in its mRNA levels, suggesting that an important level of TBP regulation occurs post-transcriptionally (*Deato and Tjian, 2007*; *D'Alessio et al., 2011*; *Liu et al., 2011*; *Zhou et al., 2013*; *Herrera et al., 2014*). We therefore set out to decipher the mechanisms by which TBP protein levels are regulated during differentiation.

Here, we show that TBP is a substrate for ubiquitination by a specific E3 ligase in vivo and in vitro. By using a combination of in vitro assays and biochemical fractionation, we identified the 480 kDa Huwe1 protein as a major E3 ligase targeting TBP. We confirmed in vitro that purified recombinant Huwe1 targets TBP for ubiquitination, while loss-of-function experiments allowed us to probe TBP regulation by the E3 ligase in vivo. We also identified USP10 as a ubiquitin-specific protease (USP) that modulates Huwe1-mediated TBP ubiquitination and protects TBP from proteasome-mediated degradation. Importantly, we were able to show that the E3 ligase becomes significantly upregulated during myotube differentiation, while its paired USP becomes dramatically downregulated. Together, our results support a model in which coordinated ubiquitination and deubiquitination activities finely balance TBP protein levels during terminal differentiation.

## Results

### Regulation of TBP protein levels by UPS during myogenesis

Consistent with several earlier observations from our lab and other studies (*Perletti et al., 1999*; *Deato and Tjian, 2007*), we found that upon myogenic differentiation of C2C12 cells TBP protein levels become significantly downregulated, while its mRNA levels remained largely unchanged (*Figure 1—figure supplement 1*), suggesting that TBP downregulation mainly occurs via modulation of its protein levels. Since the major protein degradation pathway in eukaryotic cells is mediated by the ubiquitin/proteasome system (UPS) (*Hochstrasser, 1996*; *Hershko and Ciechanover, 1998*), we treated post-mitotic myotubes with the proteasome inhibitor MG132 and checked whether TBP downregulation was affected. Strikingly, TBP protein levels were partially restored within 4 hr after MG132 treatment (*Figure 1A*), suggesting that the UPS likely plays a dominant role in regulating TBP levels during terminal differentiation.

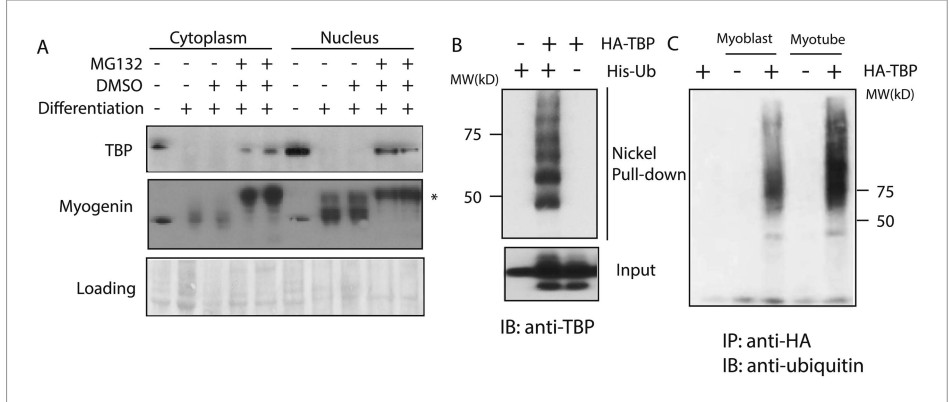

**Figure 1**. The ubiquitin-proteasome system regulates TBP protein levels. (**A**) Treatment of differentiated C2C12 myotubes with the proteasome inhibitor MG132 rescues TBP protein levels. Western blot analysis of cytoplasmic and nuclear extracts from C2C12 myoblasts and myotubes was done using antibodies against TBP and myogenin. Loading control was done by Ponceau S staining. Myotubes were treated as indicated on differentiation day 6. MG132 was used at a final concentration of 10 μM for 4 hr *, a modified version of myogenin was detected upon MG132 treatment. (**B**) TBP is ubiquitinated in vivo. 293T cells were transfected with indicated plasmids. Ubiquitin conjugates were purified using Ni-NTA resin under denatured conditions from MG132 treated cells, then subjected to western blot analysis using anti-TBP antibody. (**C**) TBP can be ubiquitinated by both C2C12 myoblast and myotube whole cell extracts in vitro. In vitro transcribed and translated HA-TBP was incubated with whole cell extracts from C2C12 myoblasts or myotubes for in vitro ubiquitination assays. After the reaction, immunopurified HA-TBP (purified using an antibody against HA-tag) was analyzed by western blot using an antibody against ubiquitin.

The following figure supplement is available for figure 1:

**Figure supplement 1**. Protein levels of TBP and TAF4 but not RNA polymerase II significantly decrease during C2C12 differentiation.

To test whether TBP is indeed a substrate for ubiquitination, we next performed in vivo ubiquitination assays, in which we treated 293T cells expressing His-ubiquitin and Homologous to the E6-AP Carboxyl Terminus (HA-TBP) with MG132 overnight. Subsequently, all ubiquitinated protein species in the cells were pulled down under denaturing conditions by using Nickel resin (See details in the 'Materials and methods'). Western blot analysis confirmed the presence of slower-migrating poly-ubiquitinated species of the TBP protein in the pull-downs, suggesting TBP is indeed ubiquitinated in cells (*Figure 1B*).

## Biochemical fractionation and identification of Huwe1 as a TBP E3 ligase

Ubiquitination is catalyzed by a cascade of three different enzymes, E1 activation enzyme, E2 conjugation enzymes and E3 ligases. The target specificity of the system is largely determined by E3 ligases, that recognize specific substrates for ubiquitination (*Nakayama and Nakayama, 2006*). To identify potential E3 ligases that specifically target TBP for ubiquitination, we used biochemical fractionation followed by an in vitro ubiquitination assay.

As a first step towards establishing an in vitro biochemical assay for TBP ubiquitination, we incubated in vitro translated HA-tagged TBP protein with either myoblast or myotube lysates supplemented with proteasome inhibitor MG132, a general deubiquitinase inhibitor (ubiquitin aldehyde), ubiquitin and an energy regenerating mix (See details in the 'Materials and methods'). After 90 min of reaction at 37°C, we purified the substrate protein by anti-HA antibody affinity chromatography and analyzed TBP ubiquitination by anti-ubiquitin western blots. Higher molecular weight ubiquitinated TBP species were detected in reactions with both myoblast and myotube lysates, suggesting TBP ubiquitination activities are present in both proliferating muscle progenitors and terminally differentiated muscle cells (*Figure 1C*).

Since it is impractical to scale up myoblast and myotube cultures to obtain sufficient materials for bulk biochemical fractionation, we then tested whether TBP can be ubiquitinated by lysates of Hela

cells. We detected significant TBP ubiquitination activities in Hela whole cell extracts (*Figure 2A*). We then separated the whole cell extract into cytoplasmic and nuclear fractions, and found that most of the TBP ubiquitination activities were present in the cytoplasmic fraction (S100) (*Figure 2A*). Using S100 as starting material, we designed a purification scheme for isolating putative TBP E3 ligases based on the in vitro ubiquitination assay. To simplify the ubiquitination assay, we used recombinant GST-tagged TBP as the substrate, which allows efficient isolation of the ubiquitinated TBP species

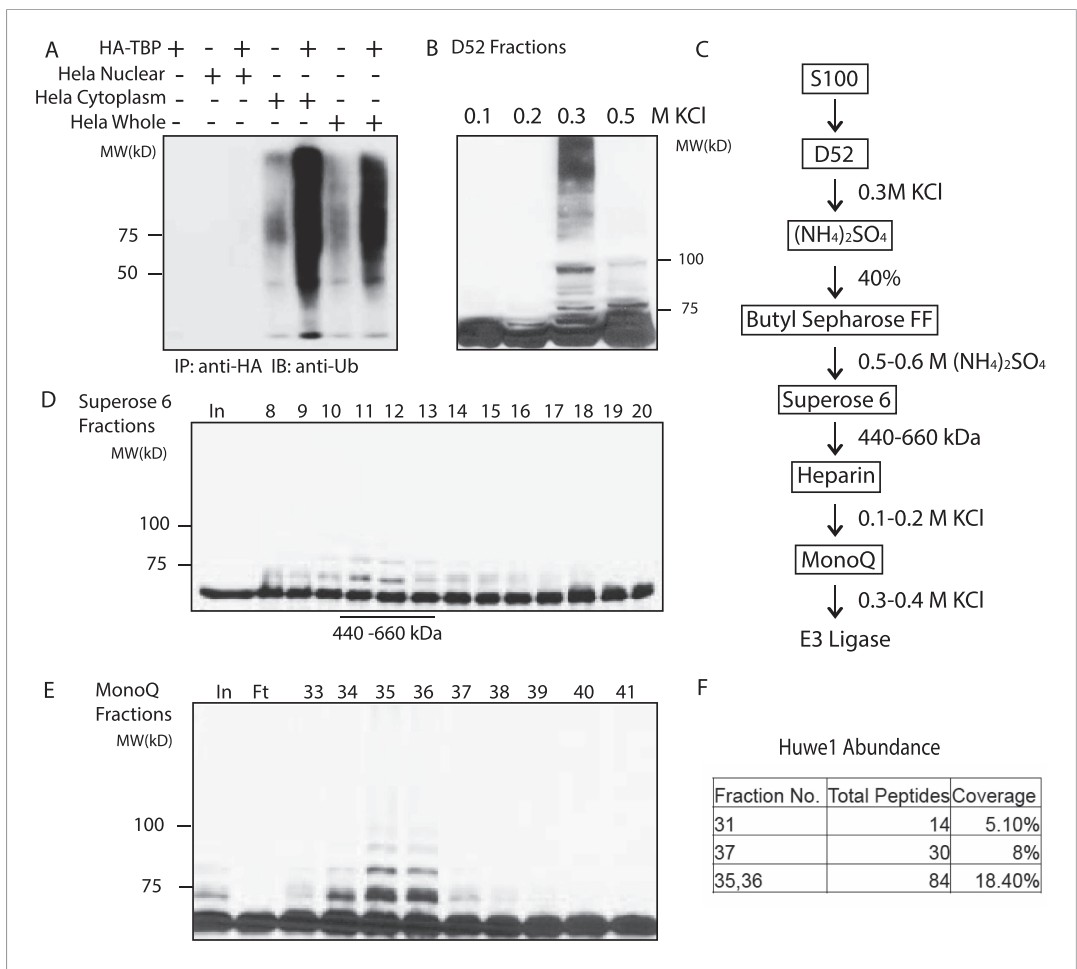

**Figure 2**. Purification of TBP E3 ligase from HeLa cytoplasmic fraction (S100). (**A**) In vitro ubiquitination of TBP by HeLa cell lysates. In vitro transcribed and translated HA-TBP was incubated with indicated HeLa cellular fractions for ubiquitination assays. After the reaction, immunopurified HA-TBP was analyzed western blot using an antibody against ubiquitin. (**B**) Fractions eluted from the ion exchange (D52) column were assayed in the presence of E1 (UBE1), E2 (UbcH5b), ubiquitin, ubiquitin aldehyde (deubiquitinase inhibitor) and MG132 (proteasome inhibitor). TBP ubiquitination was analyzed by western blots using anti-TBP antibody. (**C**) Chromatography scheme of purification of TBP E3 ligase. Hela cytoplasmic fraction (S100) was subjected to a series of chromatographic fractionation as indicated. (**D**) TBP E3 ligase migrates at a size around 440–660 kDa. Input (In) and Superose 6 fractions were assayed as in (**B**). Motilities of the peak activity (440–660 kDa) are shown (bottom). (**E**) TBP E3 ligase activity after the final Mono Q chromatography step. Reactions containing Input (In) and Mono Q fractions were performed as in (**B**). (**F**) The abundance of Huwe1 protein positively correlates with levels of TBP E3 ligase activity within selected MonoQ fractions. Protein compositions of selected MonoQ fractions were determined by Mudpit (Multidimensional Protein Identification Technology). Shown are numbers of detected peptides and percentage of coverage for Huwe1 protein in indicated fractions.

The following figure supplement is available for figure 2:

**Figure supplement 1**. GST-TBP can be used as a substrate for in vitro ubiquitination assays.

with GST resin. It is worth noting that while GST-TBP can be efficiently ubiquitinated in the reaction, we did not detect any ubiquitination of the GST protein (*Figure 2—figure supplement 1A*), suggesting that these ubiquitination activities are specific to TBP. We could also detect ubiquitinated TBP species as slower-migrating bands by anti-TBP western blots (*Figure 2—figure supplement 1B*).

To purify E3 ligases, we first fractionated the S100 lysate on an ion exchange column (D52) into multiple fractions: flowthrough, proteins bound to the column and eluted at 0.2 M KCl, 0.3 M KCl and 0.5 M KCl, respectively. None of the fractions alone was sufficient to mediate TBP ubiquitination. Reasoning that E1, E2 and E3 activities might be eluted at different salt concentrations, we then supplemented each fraction with recombinant E1 and a panel of E2s, and found that the 300 mM elution supplemented with several E2s (UbcH5 family E2s and UbcH7) robustly catalyzed TBP ubiquitination (*Figure 2B*).

We then precipitated the 300 mM D52 eluate with ammonium sulfate and further fractionate it by Butyl Sepharose, Superose 6 gel filtration, Heparin and finally MonoQ columns to enrich for E3 ligases responsible for TBP ubiquitination (*Figure 2C*) (See details in the 'Materials and methods'). Ubiquitination assays on the activities eluted from the Superose 6 sizing column suggest the molecular weight of the E3 ligase(s) to be about 440–660 kDa (*Figure 2D*). The E3 activity was eluted from MonoQ at roughly 400 mM KCl (*Figure 2E*). At this stage, the limited amount of remaining material prevented further purification so the sample was subjected to Mudpit (Multidimensional Protein Identification Technology) mass spectrometry analysis along with control fractions containing little or no detectable levels of TBP E3 activities.

Of the three E3 ligases detected by Mudpit analysis, Huwe1 emerged as the most promising candidate, as its abundance correlated very well with TBP ubiquitination activities within each fraction (*Figure 2F*). Specifically, a total of 81 Huwe1 peptides (18.4% coverage) were detected in fractions with the highest E3 activities, and this number fell to 30 and 14 peptides in the fractions with intermediate and low E3 activities, respectively. Moreover, Huwe1 protein consists of 4374 amino acids with a molecular mass of around 480 kDa (*Chen et al., 2005*; *Zhong et al., 2005*), consistent with the molecular weight predicted for the TBP E3 ligase activity after Superose 6 gel filtration (*Figure 2D*).

## Huwe1 mediates TBP ubiquitination in vitro

To test whether Huwe1 indeed mediates TBP ubiquitination, we performed in vitro ubiquitination assays with purified recombinant Huwe1 proteins (*Figure 3—figure supplement 1A*), which resulted in a dose-dependent poly-ubiquitination of TBP detected as high molecular weight species using anti-TBP antibody (*Figure 3A*). As expected, the TBP ubiquitination activity of the purified recombinant Huwe1 protein was much more robust than that of partially purified fractions, presumably due to higher Huwe1 concentrations and/or the absence of potential inhibitors of ubiquitin polymerization. When using a methyl ubiquitin that only supports monoubiquitination together with purified recombinant Huwe1 in our assay (*Hershko and Heller, 1985*), we detected a ladder of shifted bands separated by around 8 kDa similar to the TBP ubiquitination patterns generated by the partially purified Mono Q fractions (*Figure 3—figure supplement 1B*). Since these shifted bands likely correspond to different ubiquitination sites, the use of methyl ubiquitin also allowed us to estimate the number of distinct ubiquitination sites on TBP to be more than 10 (*Figure 3—figure supplement 1B*).

To confirm that this ubiquitination reaction requires direct physical interaction between TBP and Huwe1, we tested whether these two proteins coprecipitate through GST pull down experiments. Indeed, purified His-Huwe1 bound to full-length TBP protein (*Figure 3—figure supplement 2*). Moreover, this interaction is dependent on the C-terminal DNA binding domain of TBP, which is a highly conserved and highly structured region (*Figure 3—figure supplement 2*).

Previous studies showed that Huwe1 contains a HECT (Homologous to the E6-AP Carboxyl Terminus) domain, which accepts ubiquitin from an E2 ubiquitin-conjugating enzyme in the form of a thioester, and then directly transfers the ubiquitin to targeted substrates (*Rotin and Kumar, 2009*). To confirm that the HECT domain of Huwe1 is required for its TBP ubiquitination activity, we generated two mutants of Huwe1: one with the HECT domain truncated (Δ4341–4374), the other with the active site cysteine residue mutated to serine. Both mutant proteins were purified through the same procedures as the wild type protein (*Figure 3—figure supplement 1A*). Neither of the mutants was able to mediate TBP ubiquitination in vitro (*Figure 3B*, *Figure 3—figure supplement 1C*). Apparently, the N-terminal region of Huwe1 outside the catalytic domain is also required for efficient

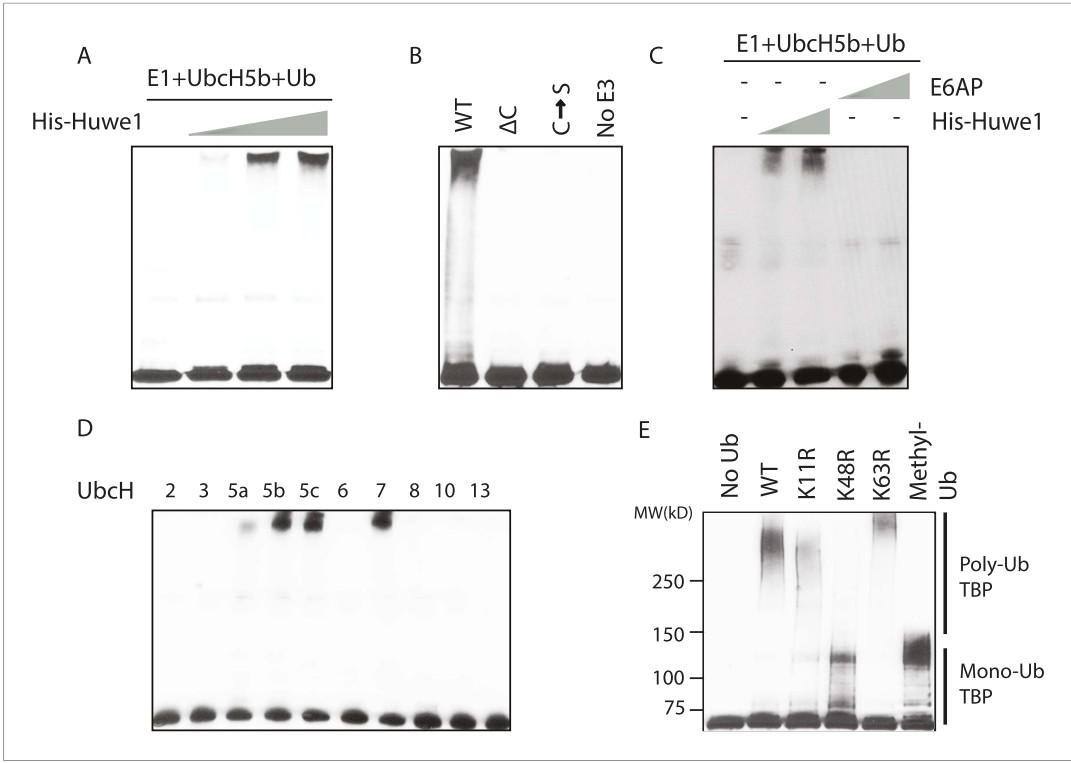

**Figure 3**. In vitro ubiquitination of TBP using recombinant Huwe1 protein. (**A**) Recombinant Huwe1 protein ubiquitinates TBP in vitro. Titrations (twofold concentration range) of recombinant His-tag Huwe1 protein are incubated with GST-TBP supplemented with E1, UbcH5b, wild-type ubiquitin and ATP regenerating mix, ubiquitin aldehyde in in vitro ubiquitination assays. After the reaction, TBP ubiquitination was analyzed through western blot using the anti-TBP antibody. (**B**) Hect domain of Huwe1 is required for its TBP ubiquitination activity. Wildtype (WT), Hect domain truncation (ΔC) and catalytic site mutant (C → S) Huwe1 were assayed as in (**A**) to test their activities to ubiquitinate TBP. (**C**) Hect domain alone is insufficient for Huwe1's TBP ubiquitination activity. Two different doses of His-tag Huwe1 (twofold range) and recombinant E6AP protein (twofold range) are used in the ubiquitination assay as in (**A**). (**D**) Huwe1 E3 activity can be supported by UbcH5 family E2s and UbcH7. A panel of different E2 conjugating enzymes is used in the in vitro ubiquitination assays. (**E**) Huwe1 mediated the K48-linked ubiquitination of TBP. Wild-type ubiquitin (WT), lysine 11 to arginine mutant (K11R), lysine48 to arginine mutant (K48R), lysine63 to arginine mutant (K63R) and lysine-methylated ubiquitin (Methyl-Ub) are used in the ubiquitination assays.
The following figure supplements are available for figure 3:

**Figure supplement 1**. Huwe1 mediates TBP ubiquitination at multiple sites.

**Figure supplement 2**. Huwe1 directly interacts with TBP in vitro.

TBP ubiquitination, as the homologous HECT domain from E3 ligase E6AP was unable to carry out TBP ubiquitination (*Figure 3C*, *Figure 3—figure supplement 1D*).

Having established that Huwe1 mediates TBP ubiquitination in vitro, we then asked which E2 conjugating enzymes work best with Huwe1 in this reaction. Many recent studies have revealed the importance of E2 conjugating enzymes in determining the length and topology of ubiquitin chains and hence the cellular consequences of ubiquitination (*Ye and Rape, 2009*). We therefore tested the ability of a panel of E2s to collaborate with Huwe1 in mediating TBP ubiquitination, and found that both the UbcH5 family E2s and UbcH7 are active in the in vitro ubiquitination assay with recombinant purified Huwe1 (*Figure 3D*, *Figure 3—figure supplement 1E*), consistent with the E2 enzymes that worked with the crude 0.3 M KCl fraction from the D52 column.

We also asked whether TBP ubiquitination has specific poly-ubiquitin chain topologies. It was shown that poly-ubiquitin chains on a substrate can be formed using any of the seven lysines in ubiquitin, and

specific ubiquitin chain topologies are linked to distinct biological pathways (*Komander and Rape, 2012*). To map the topology, we performed in vitro TBP ubiquitination assays using ubiquitin mutants with individual lysines mutated to arginines. Strikingly, when K48R ubiquitin was used, poly-ubiquitinated TBP species were significantly diminished (*Figure 3E*, lane 4), suggesting that Huwe1 likely mediates K48-linked ubiquitin chain formation on TBP. Ubiquitin chains of this topology usually lead to proteasome-mediated degradation (*Komander and Rape, 2012*).

## Huwe1 regulates TBP protein levels in proliferating cells

Having established that Huwe1 supports TBP K48 linkage ubiquitination in vitro, we next tested whether Huwe1 regulates TBP protein levels in vivo. Knocking-down Huwe1 in 293T cells with two different shRNAs resulted in increased levels of endogenous TBP protein (*Figure 4A*). To determine relative TBP protein stability more quantitatively in cells, we developed an in vivo TBP half-life measurement system based on a fluorescence timer assay (*Khmelinskii et al., 2012*). In this assay, TBP is fused to both a monomeric red fluorescent protein (mCherry), which takes hours to complete its maturation, and a monomeric superfolder green fluorescent protein (sfGFP), which becomes fluorescent within minutes after synthesis. Due to the slow maturation rate of mCherry relative to sfGFP, the mCherry/sfGFP intensity ratio at steady state correlates with the half-life of the fusion protein, independent of the protein synthesis rate (*Figure 4—figure supplement 1A*). We co-transfected 293T cells with plasmids expressing Huwe1 shRNAs and TBP-mCherry-sfGFP fusion protein. Knockdown of Huwe1 by 2 independent shRNAs led to significant increases in mCherry/GFP intensity ratios, evident in both imaging and flow cytometry analysis (*Figure 4B*, *Figure 4—figure supplement 1B*). Similarly, lentivirus-mediated Huwe1 shRNA knockdown in C2C12 myoblasts leads to elevated endogenous TBP protein levels (*Figure 5A*). We further tested how Huwe1 affects TBP protein half-life by cycloheximide (CHX) chase experiments. Consistent with results from fluorescent timer experiments, Huwe1 knockdown by two independent shRNAs both increased TBP protein half-life in myoblasts (*Figure 5—figure supplement 1*). Taken together, our results strongly suggest that Huwe1 modulates TBP protein levels in living cells.

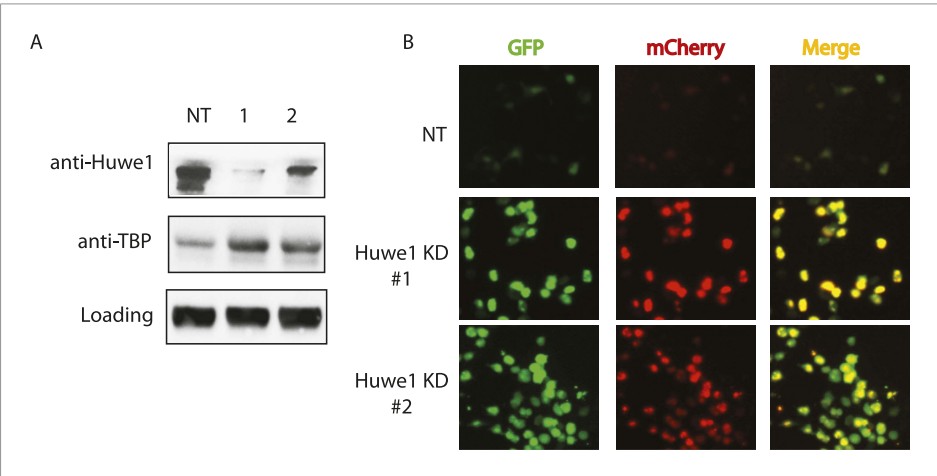

**Figure 4**. Huwe1 regulates TBP protein levels in proliferating cells. (**A**) Huwe1 knockdown increases TBP protein levels in 293T cells. Western blot analysis of whole cell extracts from control (NT) and Huwe1 knockdown 293T cells was done using antibodies against Huwe1 and TBP. In this assay, Tubulin signals were used as loading controls. (**B**) Huwe1 knockdown increases protein half-life of TBP-fluorescent-timer fusion protein. Fluorescent images of 293T cells cotransfected with a plasmid expressing the TBP-fluorescent-timer fusion protein and control pLKO.1 (NT) or two different pLKO.1 plasmids expressing shRNA against Huwe1 (Huwe1 KD#1 and Huwe1 KD#2) are taken using the same exposure time, then processed in parallel with ImageJ.

The following figure supplement is available for figure 4:

**Figure supplement 1**. Huwe1 knockdown prolongs half-life of fluorescent-timer TBP fusion protein in 273T cells as measured by fluoresent-timer assays.

**Figure 5**. Huwe1 regulates TBP protein level in myoblasts and myotubes. (**A**) Lentivirus-mediated depletion of Huwe1 elevates TBP protein level in C2C12 myotubes. Cytoplasmic and nuclear extracts of C2C12 myoblasts infected with control (NT) lentiviruses or two different lentiviruses targeting Huwe1 (1, 2) were analyzed by western blots using antibodies against Huwe1 and TBP. Signals in Ponceus S staining were used as loading controls. (**B**) Up-regulation of Huwe1 protein levels during C2C12 differentiation. Cytoplasm and nuclear extracts of proliferating C2C12 myoblasts (D0) and differentiated C2C12 myotubes (D6) were analyzed by western blots using antibodies against Huwe1 and TBP. Signals in Ponceus S staining were used as loading controls. Results from two biological replicates are shown. (**C**) Huwe1 knockdown in differentiated myotubes partly rescues TBP protein levels. Cytoplasmic and nuclear extracts of differentiation day 8 myotubes infected with either control lentiviruses (NT) or two different lentiviruses targeting Huwe1 (1,2) were analyzed by western blots using antibodies against Huwe1 and TBP. Signals in Ponceus S staining were used as loading controls.

The following figure supplement is available for figure 5:

**Figure supplement 1**. Huwe1 knockdown increases TBP protein half-life in C2C12 myoblasts.

## Upregulation of Huwe1 during myogenesis promotes TBP destruction

Although our results indicated that Huwe1 regulates TBP protein levels in vivo, it remained unclear what causes the dramatic downregulation of TBP protein levels upon myogenesis. One possibility is that Huwe1 E3 ligase activities are upregulated during muscle differentiation. To test this hypothesis, we examined Huwe1 protein levels in myoblasts and myotubes. Strikingly, in contrast to dramatic decreases in TBP protein levels, Huwe1 protein levels are significantly upregulated in myotubes (*Figure 5B*). This observation is consistent with previously published results showing that Huwe1 transcripts are enriched in muscle tissues (*Schwarz et al., 1998*; *Chen et al., 2005*). To confirm that Huwe1 is indeed responsible for active TBP degradation in myotubes, we treated myotubes with high titer viruses expressing shRNAs against Huwe1 on differentiation Day 4. This treatment resulted in abnormal morphologies of the polynucleated myotubes (*Figure 6D*), accompanied by increased TBP protein levels (*Figure 5C*). These results support a model wherein the upregulation of Huwe1 protein levels during myogenesis likely leads to enhanced TBP ubiquitination and subsequent degradation of TBP protein.

## Huwe1 is required for muscle differentiation and maintenance of normal muscle morphologies

Based on the observations that Huwe1 regulates TBP protein levels in both proliferating and differentiated muscle cells and that Huwe1 protein levels increase during myogenesis, we investigated the effects of Huwe1 knockdown on C2C12 myoblast proliferation and differentiation. When cultured in proliferation medium, the Huwe1 knockdown cells are morphologically similar to control cells (*Figure 6—figure supplement 1A*). Consistent with this observation, genome-wide transcriptome analysis revealed no significant gene expression changes for cell-cycle and house-keeping genes upon Huwe1 knockdown in C2C12 cells (*Figure 6—figure supplement 1B*; *Figure 6—source data 3*). However, it is worth noting that two important myogenic regulator genes, *Myod1* and *Mef2c*, showed a ~twofold decrease in expression levels in Huwe1 knockdown cells (*Figure 6—figure supplement 1B*). We next performed myogenic differentiation experiments to evaluate the lineage commitment potential of Huwe1 depleted myoblasts. Interestingly, Huwe1 knockdown cells failed to differentiate

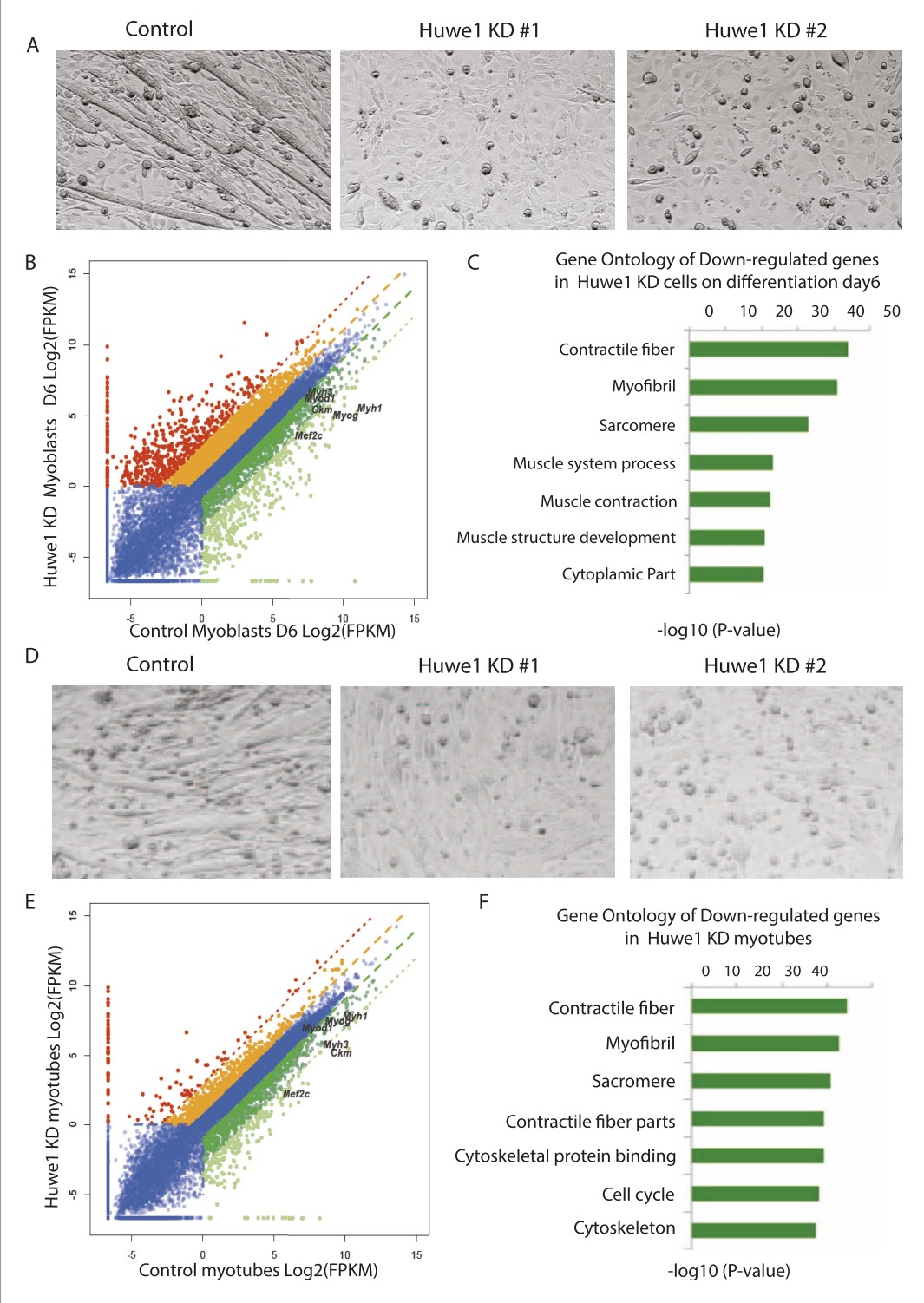

**Figure 6**. Huwe1 is required for muscle differentiation and maintenance of normal muscle morphologies. (**A**) Huwe1 knockdown impaired C2C12 differentiation efficiency. Shown are phase contrast images of control and Huwe1 knockdown C2C12 myoblasts after 6 days of differentiation in 2% horse serum. (**B**) Muscle marker genes were not efficiently induced in Huwe1 knockdown C2C12 cells after 6 days of differentiation. Differential gene expression analysis was used to characterize the effects of Huwe1 knockdown on muscle differentiation using mRNA-seq data. The log2 scale values of fragments per kilobase per million (FPKM) for genes in both control and Huwe1 knockdown samples are plotted in the graph. Dark green, twofold down; light green, eightfold down; orange, twofold up; red,

*Figure 6. continued on next page*

*Figure 6. Continued*

eightfold up. (**C**) Goseq gene ontology analysis of genes down-regulated in Huwe1 knockdown samples. The list shows top categories ranked by p-value. (**D**) Knockdown of Huwe1 in differentiated muscle cells affected normal muscle morphology. Shown are phase contrast images of control and Huwe1 knockdown C2C12 myotubes. (**E**) Huwe1 knockdown in terminally differentiated myotubes reduced transcription levels of muscle marker genes. Differential gene expression analysis to characterize the effects of Huwe1 knockdown in terminally differentiated muscle cells using mRNA-seq data. Figure was labelled the same way as (**B**). (**F**) Goseq gene ontology analysis of genes down regulated by Huwe1 knockdown in myotubes. List showes top categories ranked by p-value.

The following source data and figure supplements are available for figure 6:

**Source data 1**. Differentiation gene expression analysis of control and Huwe1 knockdown C2C12 cells on differentiation Day 6 by RNA-seq.

**Source data 2**. Differentiation gene expression analysis of control and Huwe1 knockdown myotubes on differentiation Day 8 by RNA-seq.

**Source data 3**. Differentiation gene expression analysis of control and Huwe1 knockdown C2C12 cells on differentiation Day 0 by RNA-seq.

**Figure supplement 1**. Huwe1 knockdown causes modest changes in the transcriptional program of C2C12 myoblasts.

**Figure supplement 2**. Huwe1 knockdown inhibits the induction of master myogenic factors during C2C12 differentiation.

into polynucleated myotubes efficiently even after 6 days of differentiation (*Figure 6A*). Consistent with this phenotypic observation, the induction of several master myogenic regulators (*Myod1*, *Myog* and *Mef2c*) was inhibited in the Huwe1 knockdown cells during differentiation (*Figure 6—figure supplement 2*). To gain more information about potential changes in transcription between control and Huwe1 knockdown cell populations, we compared gene expression profiles of these two cell types by RNA-seq analysis. Differential gene expression followed by gene ontology analysis revealed that many genes critical for muscle development have lower expression levels in Huwe1 knockdown cells (*e.g.*, *Myod1*, *Myog*, *Mef2a*, *Mef2c*, *Ckm* and *Myh3*) on differentiation Day 6 (*Figure 6B,C*; *Figure 6—source data 1*), suggesting the functional importance of Huwe1 during in vitro muscle differentiation. We further tested this notion by directly knocking down Huwe1 in differentiated myotubes. Loss of Huwe1 induced abnormal morphological changes of myotubes (*Figure 6D*). Consistent with these morphological changes, differential gene expression followed by gene ontology analysis revealed that many genes (*e.g.*, *Myog*, *Mef2a*, *Mef2c*, *Ckm* and *Myh3*) important for muscle development are downregulated in Huwe1 depleted myotubes (*Figure 6E,F*; *Figure 6—source data 2*).

## Deubiquitinase USP10 protects TBP from degradation, and downregulation of USP10 is important for myogenesis

In C2C12 myoblasts, TBP protein is present at substantial levels despite the presence of Huwe1 E3 ligase. We therefore reasoned that there must be activities in these cells that may be antagonizing the ubiquitination of TBP by Huwe1 and thereby protect it from degradation. Interestingly, a previous study reported that in *Saccharomyces cerevisiae*, Ubp3 deubiquitinates Tbp1 both in vivo and in vitro and prevents Tbp1 from proteasome-mediated degradation (*Chew et al., 2010*). A protein sequence homology comparison suggests that ubiquitin-specific protease 10 (USP10) is the likely mammalian homolog of Ubp3 (*Cohen et al., 2003*). To test whether USP10 deubiquitinates and stabilizes TBP in mammalian cells, we overexpressed USP10 in 293T cells and monitored TBP ubiquitination levels in these cells. Strikingly, overexpression of USP10 significantly inhibited TBP ubiquitination (*Figure 7A*). To test whether USP10 deubiquitinates TBP through direct protein–protein interactions, we used HA-TBP to co-IP with Flag-USP10 co-expressed in 293T cells. Our experiment revealed that TBP can co-immunoprecipitate with USP10 (*Figure 7B*). Importantly, both western blot and flow cytometry

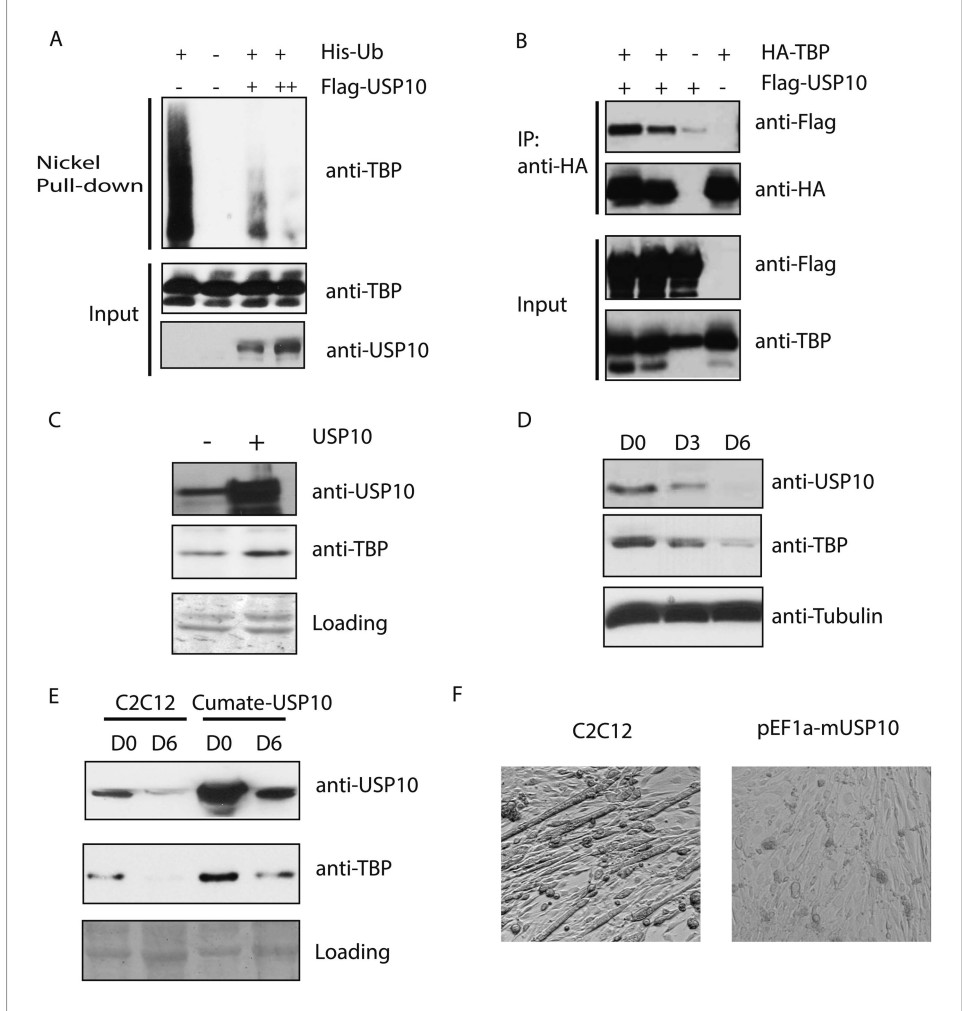

**Figure 7**. Ubiquitin-specific protease 10 (USP10) regulates TBP protein level during myogenesis. (**A**) USP10 overexpression inhibits TBP ubiquitination in vivo. 293T cells were transfected with indicated plasmids as well as a plasmid expressing HA-TBP. Ubiquitin conjugates were purified using Ni-NTA resin under denatured conditions from MG132 treated cells, then subjected to western blot analysis using anti-TBP antibody. Input lysates (1%) are analyzed using antibodies against TBP and USP10. (**B**) USP10 interacts with TBP. Coimmunoprecipation using antibodies against HA epitope was performed in 293T cell lysates transfected with indicated plasmids. Input (5%) and precipitated proteins were analyzed using antibodies as indicated. (**C**) USP10 overexpression results in modest increases of TBP protein levels in C2C12 myoblasts. Western blot analysis of whole cell extracts from control or USP10 overexpressing C2C12 myoblasts (stable cell line) were done using antibodies against USP10 and TBP. Signals in Ponceau S staining were used as loading controls. (**D**) USP10 protein level decreases during C2C12 differentiation. Whole cell extracts of myoblasts (D0) and myotubes on differentiation Day 3 (D3) and Day 6 (D6) were analyzed by western blots using antibodies against USP10, TBP and tubulin. (**E**) Induction of USP10 rescues TBP protein levels in differentiated myotubes. Whole cell extracts from myoblasts (D0) and myotubes (D6) of control and *Usp10* inducible (Cumate-USP10) C2C12 cells incubated with 10X cumate solution were analyzed by western blot using antibodies against USP10 and TBP. Signals in Ponceau S staining were used as loading controls. (**F**) USP10 overexpression inhibits C2C12 myoblasts differentiation. Shown are phase contrast images of control and USP10 overexpressing C2C12 myoblasts after 6 days of differentiation.

The following source data and figure supplements are available for figure 7:

**Source data 1**.

**Figure supplement 1**. USP10 overexpression elevates TBP protein levels and impairs differentiation capacity.

**Figure supplement 2**. USP10 overexpression increases TBP protein half-life in C2C12 myoblasts.

analysis confirmed modestly increased TBP protein levels in a C2C12 stable cell line overexpressing USP10 (*Figure 7C*, *Figure 7—figure supplement 1A*). Moreover, cycloheximide chase experiments (CHX) revealed that USP10 overexpression increased TBP protein half-life in C2C12 myoblasts (*Figure 7—figure supplement 2*) . These results suggest that USP10-mediated TBP deubiquitination might also contribute to regulating TBP levels during myogenesis. To further test this possibility, we compared the protein levels of USP10 before and after myotube differentiation. Interestingly, similar to TBP, the protein levels of USP10 decrease during in vitro muscle differentiation (*Figure 7D*). We then examined whether increasing USP10 protein levels in post mitotic myotubes can rescue TBP protein levels. To do this, we generated a stable cell line with an inducible USP10 transgene. Upon addition of the inducer (cumate), USP10 can be induced to higher levels in both myoblasts and myotubes (*Figure 7E*). USP10 overexpression in myotubes resulted in higher levels of TBP protein (*Figure 7E*). To probe whether UPS10 downregulation during myogenesis is of functional significance, we evaluated the differentiation potential of the C2C12 cell line overexpressing USP10. These cells failed to differentiate into polynucleated myotubes in differentiation medium (*Figure 7F*). Consistent with this observation, transcriptome analysis revealed that several important myogenic markers (e.g., *Myod1*, *Myog*, *Myh3* and *Ckm*) failed to be induced in cells overexpressing USP10 even on differentiation Day 6 (*Figure 7—figure supplement 1B*). Taken together, our results suggest that a combination of Huwe1 upregulation and USP10 downregulation may be important for modulating TBP degradation during myogenesis.

## Discussion

### Regulation of TBP protein levels by the ubiquitin-proteasome system

Given its central role in transcription regulated by all three classes of RNA polymerases, TBP has been long considered to be a universally required component of cellular transcription. However, recent studies have shown that the protein levels of TBP as well as other canonical components of the TFIID complex can become significantly down-regulated in terminally differentiated myocytes, hepatocytes and adipocytes (*Deato and Tjian, 2007*; *D'Alessio et al., 2011*; *Zhou et al., 2013*; *Herrera et al., 2014*). Moreover, decreases in TBP protein levels are much more pronounced than changes in its mRNA levels. However, the mechanism for regulating cellular TBP protein levels remained largely unknown. Here, we show that the ubiquitin-proteasome system plays a dominant role in mediating TBP protein levels in terminally differentiated myotubes. Interestingly, we found that an E3 ligase activity responsible for ubiquitinating TBP is also present in proliferating cell types such as myoblasts and Hela cells, where TBP is highly expressed. These results suggest that cellular TBP protein levels are under active surveillance by the ubiquitin-proteasome system and that TBP protein levels may be tightly regulated in a cell-type specific manner.

### Huwe1 and USP10 control TBP ubiquitination states

Using a combination of biochemical fractionation and in vitro ubiquitination assays, we identified the Hect domain containing Huwe1 as a key E3 ligase capable of mediating TBP ubiquitination in vitro (*Figure 3*). These in vitro biochemical studies subsequently led us to a series of experiments confirming that Huwe1 indeed also regulates TBP levels during myogenesis of C2C12 cells (*Figure 5*).

Huwe1 is essential for animal development. Specifically, deletion of *Huwe1* results in embryonic lethality, with knock-out embryos displaying hemorrhage in the abdominal region by embryonic day 14.5 (E14.5), followed by growth impairment, necrosis, and eventual death (*Kon et al., 2012*). In addition to TBP, Huwe1 has also been suggested to target several other proteins involved in cell-cycle check point and apoptosis, including p53, MCL-1, N-MYC, C-MYC and CDC6. Interestingly, although Huwe1 mRNA is ubiquitously expressed in various tissues, it is particularly enriched in skeletal muscle, which is the tissue where a dramatic decrease in TBP protein levels during terminal differentiation was first reported (*Schwarz et al., 1998*; *Chen et al., 2005*; *Deato and Tjian, 2007*). Consistent with this observation, we found that Huwe1 is significantly upregulated during in vitro muscle differentiation of C2C12 cells and that up-regulation of this E3 ligase appears to be functionally important for myogenesis and maintenance of normal muscle morphology (*Figure 6*).

As part of the TBP surveillance system, we also found that a deubiquitinase, UPS10, contributes to the regulation of TBP ubiquitination and degradation by counteracting the Huwe1 E3 ligase activity. USP10 is a ubiquitously expressed deubiquitinase, whose substrates include tumor suppressor p53 (*Yuan et al., 2010*). The exact role of USP10 during development remains unclear due to the absence

of mouse models. However, we found that a stable cell line (C2C12) overexpressing USP10 is impaired in myotube formation, suggesting that down-regulation of USP10 may also be a prerequisite for efficient differentiation of myoblasts into myotubes in culture. Deubiquitinases achieve their target specificities through either direct recognition of their substrates or targeting specific ubiquitin chain topologies. Our immune-precipitation experiments suggest that USP10 can recognize ubiquitinated TBP through direct protein–protein interactions (*Figure 7B*). It remains unclear at this point whether there are other deubiquitinases that can also recognize ubiquitinated TBP.

It is worth noting that due to the promiscuous 'one-to-many' relationship between E3 ligase, Deubiqutinase (DUBs) and their substrates, it is difficult to directly test whether the myogenic defects we observed after the loss of Huwe1 or USP10 over-expression are directly due to the failure of down-regulating TBP during differentiation or some other consequences of depleting an E3 ligase or over-expressing a deubiquitinase. However, given the seminal role of TFIID/TBP in promoting the transcription of cell cycle and DNA replication genes (*Um et al., 2001*), it is reasonable to speculate that down-regulation of TBP should at least influence cell cycle exit of myoblasts, a key step during myotube differentiation. In the future, it may be interesting to study the functional role of Huwe1 and UPS10 during muscle development in vivo and how these two enzymes may regulate TBP protein levels in mouse models. Since TBP protein levels also become dramatically reduced in terminally differentiated hepatocytes and adipocytes (*D'Alessio et al., 2011*; *Zhou et al., 2013*), it will also be interesting to test whether Huwe1 and UPS10 contribute to TBP downregulation in these other cell types. It also remains unclear whether there are other E3s that would work together with Huwe1 to facilitate TBP protein degradation in terminally differentiated muscle cells. In addition to TBP, other components of the TFIID complex also become down-regulated during terminal differentiation, and in the future it will be worth investigating whether they are targeted by the same or different E3/deubiquitinase pairs.

## Fine-tuning of TBP protein levels during muscle differentiation

Our results suggest that significant up-regulation of Huwe1 and simultaneous down-regulation of UPS10 during myotube differentiation (*Figure 5B*, *Figure 7D*) may play an important role in regulating proper TBP protein levels during muscle differentiation (*Figure 8*). One striking observation is that although the protein levels of TBP and TAFs are significantly down-regulated, the protein levels of other basal transcription factors like RNA polymerase II remain largely unchanged (*Figure 1—figure supplement 1*). We still have not fully sorted out the functional importance of this selective down-regulation of TBP and TAFs during terminal differentiation.

In several previous studies, we observed that in addition to the loss or depletion of TBP during terminal differentiation of cell-types including myotubes and adipocytes, one of the so-called 'orphan TAFs' such as TAF3 and TAF7l, respectively, becomes up-regulated or remains highly expressed while the other prototypic TFIID subunits become down-regulated (*Deato and Tjian, 2007*; *Yao et al., 2011*; *Zhou et al., 2013*). In the case of myoblast to myotube formation, we also detected the presence of a TBP related factor, TRF3, that appeared to play some role in possibly either substituting for TBP, as seen by in vitro reactions, or otherwise remaining detectable during terminal differentiation (*Deato and Tjian, 2007*; *Deato et al., 2008*). This led us, initially, to propose that the loss of TBP might be accompanied by the assembly of different initiation complexes lacking TBP but containing TRFs and orphan TAFs. However, in several subsequent studies including ESC's, adipocytes and motor neurons, although we consistently observed the emergence and up-regulation of orphan TAFs (i.e. TAF3, TAF7l and TAF9b), we did not detect any up-regulation of TRFs (*Liu et al., 2011*; *Zhou et al., 2013*; *Herrera et al., 2014*). Instead, in each of these cases, ChIP-seq experiments suggested that these orphan TAFs were associated both with core promoters (i.e. TSS) as well as either distal or proximal enhancers of target genes (*Liu et al., 2011*; *Zhou et al., 2013*; *Herrera et al., 2014*). It was also reported that KO mice lacking TRF3 showed no obvious muscle development phenotypes casting doubt on the importance of TRF3 in muscle development in vivo (*Gazdag et al., 2009*). It is therefore possible that both the loss of TBP and TFIID components in certain cell-types is not necessarily accompanied by a change in the core promoter recognition complex as initially postulated (i.e. substituting various TBP/TAFs/TRFs) but a more elaborate and as yet unclear mechanism may come into play potentially involving new interactions between certain TAFs and cell-type specific transcription factors bound to enhancers. In any case, in all our studies of terminal differentiating cell types, we observed a significant, sometimes dramatic loss of TBP and TFIID components that is often accompanied by persistent or differential up-regulation of an orphan TAF.

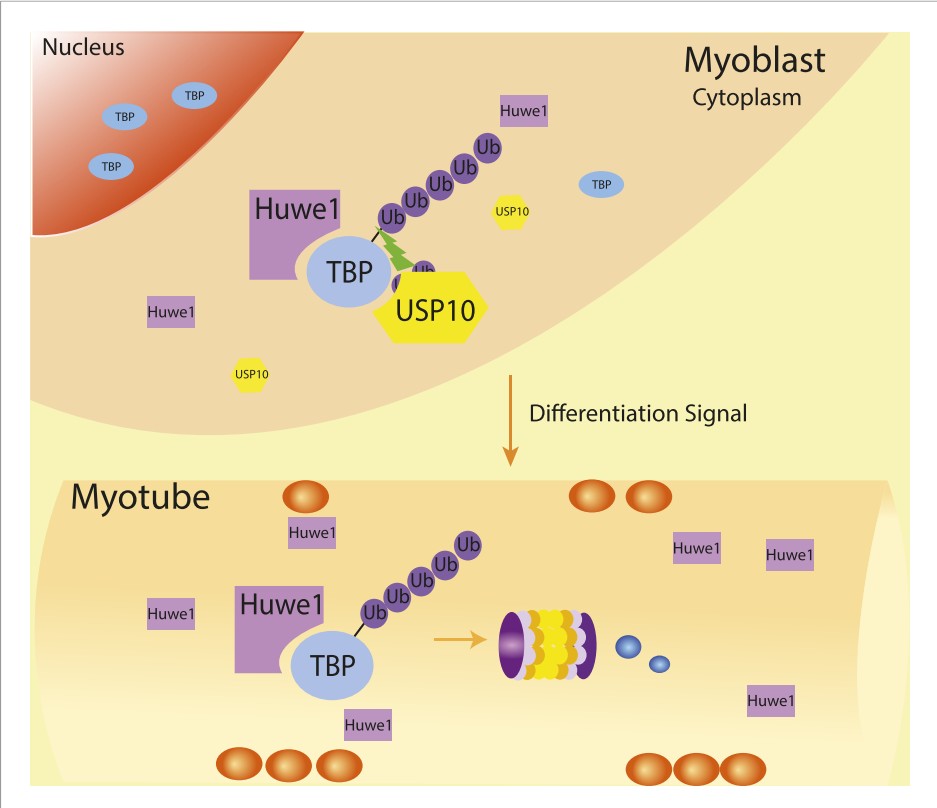

**Figure 8**. Schematic representation of coordinated regulation of TBP protein level in proliferating and differentiated cells by E3 ligase Huwe1 and deubiquitinase USP10. Both Huwe1 E3 ligase and USP10 deubiquitinase are present in proliferating myoblasts to maintain steady state TBP protein levels. Upon stimulation by differentiation signals, Huwe1 protein levels increase while USP10 protein levels decrease, resulting in increased TBP ubiquitination and degradation by the proteasome.

An interesting possibility is that down-regulation of TBP protein levels could differentially affect genes with distinct TBP-promoter binding affinities. Recent single molecule studies suggest that transcription factor temporal occupancies at high affinity sites are less sensitive to changes in concentration than those at low-affinity sites (*Chen et al., 2014*). Thus, it is possible that, by fine-tuning TBP protein levels, genes with weak TBP/TFIID binding promoters become selectively turned down while genes with strong TBP/TFIID binding promoters remain largely unaffected thereby executing another mechanism for differential gene regulation.

## Materials and methods

### C2C12 culture and differentiation

C2C12 myoblasts were obtained from ATCC (CRL-1772, Manassas, VA), and cultured in DMEM medium (11995-065, Life technologies, Frederick, MD) containing 10% FBS (26140-079, Life technologies) at sub-confluent densities (<70%). For differentiation, C2C12 cells were grown to 100% confluence and shifted to differentiation medium DMEM (11995-065, Life technologies) containing 2% horse serum (16050-130, Life technologies). Since decreases in protein levels of TBP and TAFs are generally more dramatic starting from differentiation day 6, we collected myotubes on this day for most analysis unless otherwise noted.

### Protein extraction preparation

Cytoplasmic and Nuclear extraction were prepared as previously described (*Dignam et al., 1983*). For whole cell extraction preparation, cell pellets were incubated in lysis buffer (1% NP-40, 150 mM NaCl, 50 mM, pH 8.0 Tris–HCl, 0.5% sodium deoxycholate, protease inhibitor cocktails (05892791001,

Roche, Nutley, NJ)) for 30 min on ice, passed through needles (20 Gauge) 7–8 times, then centrifuged at 15,000 rpm for 30 min. Supernatant was kept as the cell lysate. Protein concentration of soluble cell lysate was measure by Bradford assay.

## Antibody information

Monoclonal antibodies against TBP (ab61411), Huwe1 (ab78397) and polyclonal antibodies against USP10 (ab72486), RNA polymerase II (ab52202), HA-tag (ab9110) from abcam (Cambridge, MA); polyclonal antibody against ubiquitin (07–375) from EMD MILLIPORE (Billerica, MA); monoclonal antibodies against TAF4 (612054, BD Transduction Laboratories) and Myogenin (556358, BD Pharmingen) from BD Biosciences (Franklin Lakes, NJ); anti-Flag tag (M2) monoclonal antibody from Sigma Aldrich (St. Louis, MO).

## In vivo ubiquitination assay

293T cells were transfected with $His_6$-ubiquitin (6 µg) and HA-TBP (2 µg) as indicated in *Figure 1B* and *Figure 7A* by calcium phosphate transfection. 24 hr after transfection, cells with treated with 1 µM MG132 overnight to accumulate ubiquitinated proteins. Cells were harvested for in vivo ubiquitination assays, which were performed as described previously (*Jin et al., 2012*). Western blots were performed using anti-TBP antibody.

## In vitro ubiquitination assay

In a 30-µl reaction, 2 µl of in vitro translated HA-TBP (generated using TNT SP6 Coupled Reticulocyte Lysate System (L4600, Promega, Madison, WI)) or 200 ng of recombinant GST-TBP was incubated with an ATP regenerating system (37.5 mM creatine phosphate, 5 mM ATP, pH 8.0, 5 mM $MgCl_2$). 2 µg of ubiquitin (U-100, Boston Biochem, Cambridge, MA), 50 ng E1 (E-305, Boston Biochem), 100 ng UbcH5b (E2-622, Boston Biochem), 2 µM ubiquitin aldehyde (U-201, Boston Biochem), 10 µM MG132 (474790, EMD Millipore) and 10 µg of S100, partial purified fractions or purified recombinant Huwe1 protein for one and a half hour. For detection of ubiquitinated species using the anti-ubiquitin antibody, after the reaction, reaction mixtures were diluted in buffer (1% NP40, 150 mM NaCl, 50 mM Tris–HCl pH 7.5, 0.5% sodium deoxycholate), followed by immunoprecipitation using anti-HA antibody/protein G sepharose beads or pull-down using glutathione sepharose beads. Purified TBP proteins were then subjected to three washes with the same buffer, eluted by boiling in 2×SDS loading buffer and analyzed by western blots. For detection of ubiquitinated species using the anti-TBP antibody, reactions were terminated with 2×SDS sample buffer, and then analyzed by western blots.

## Purification of TBP E3 ligase

All steps were performed at 4°C. Cytoplasmic extracts were prepared from 200 L of Hela cells using Buffer A (10 mM HEPES pH7.9, 0.5 mM MgCl2, 100 mM KCl, 0.5 mM DTT), then applied to a D52 DEAE cellulose (Whatman) column, washed extensively at 100 mM KCl, 200 mM KCl and eluted at 300 mM KCl. This 300 mM KCl fraction was then precipitated with ammonium sulfate (40% saturation), and re-suspended in Buffer A containing 100 mM KCl and 800 mM $(NH_4)_2SO_4$ . The soluble fraction was applied to Hitrap Butyl Sepharose FF hydrophobic column (17-5197-01, GE Healthcare Life Sciences, Pittsburgh, PA), subjected to 10 column volume (CV) washes using Buffer A containing 800 mM $(NH_4)_2SO_4$, eluted with a 10 CV linear gradient from Buffer A containing and 800 mM $(NH_4)_2SO_4$ to Buffer A. Active fractions were then pooled and separated on a Superose6 10/300 GL (17-5172-01, GE Healthcare Life Sciences) equilibrated with Buffer A. Active Superose 6 fractions with an approximate molecular mass of 440–660 kDa were pooled and supplemented with 0.1 mg/ml insulin (11376497001, Roche). Pooled fractions were applied to a Hitrap Heparin HP column (17-0406-01, GE Healthcare Life Sciences), washed with 10 CV of Buffer A, and eluted with a 10 CV linear gradient from 0.1M KCl to 0.5 M KCl. Active Heparin fractions were pooled and dialyzed against Buffer A. Dialyzed fractions were then applied to a Mono Q GL column (17-5166-01, GE Healthcare Life Sciences), washed with 10 CV of Buffer A, then developed with a 20 CV linear gradient from 100 mM KCl to 500 mM KCl. Active TBP E3 ligase fractions eluted from 300 mM to 400 mM KCl.

## Plamids for Recombinant Huwe1 protein production

pFastBac plasmid for his-tag wild-type Huwe1 expression was generously provided by Dr. Qing Zhong (UTSW). Huwe1 catalytic domain deletion was created by replacing the a ApaI-NotI fragment from

pCI-neo wild-type Huwe1 plamids (provided by Dr. Qing Zhong(UTSW)) with a ApaI-NotI fragment containing Huwe1 deletion (amplified using primers 5′-CGGGGTCGGGCCCGCCTCCTGGTAGGCAAC-3′ and 5′-ACGATGCGGCCGCTTATGTGTGAGCTGAAGGCAGGCG-3′). The full-length Huwe1 cDNA with C-terminal deletion was then cloned into pFastBac as a SalI-NotI fragment. pFastbac plasmid for Huwe1 catalytic site mutant was created in a similar way, except for that a different reverse primer (5′-CGG GGTCGGGCCCGCCTTAGGCCAGCCCAAAGCCTTCAGAGC ACTCCTGGATAGCCAACAGTAGCATG TGGCGGAGCTTCTCAAAGCTCTCATAGGCAGGCAGATCCAGCTGATTAAAACTTGTGTGAGCTGAA-3′) was used for ApaI-NotI fragment generation.

## Purification of WT and mutant recombinant mule protein

pFastBac plasmids were transfected into Sf9 using Cellfectin II reagent (10362-100, Life technologies) following manufacture's instructions. P3 viruses were used to infect Sf9 cells for protein production. The cells were harvested 48 hr post-infection. Cell pellets were re-suspended in Buffer A (50 ml Buffer A for cell pellets from 1 L culture), and homogenized with a glass douncer. Cytoplasmic fraction was collected by centrifuging at 10,000 rpm for 30 min, added in imidazole to final concentration of 10 mM, and then loaded onto a Ni-NTA gravity column. The column was washed with 20 CV Buffer A containing 10 mM imidazole, and the bound His- Huwe1 was eluted using Buffer A containing 100 mM imidazole. A Superdex200 Gl gel filtration column was used to further purify the His-Mule elution. PageBlue staining was used to check Huwe1 protein in each fraction. Glycerol was added to the fractions containing Huwe1 to a final concentration of 5%, and the proteins were flash frozen in liquid nitrogen and stored at −80℃.

## shRNA-mediated knockdown of Huwe1 in C2C12 myoblasts and myotubes

Huwe1 knockdown in C2C12 myoblasts and myotubes were performed using lentiviruses. Viruses were produced by transfecting pLKO.1 shRNA plasmids targeting Huwe1 (targeting sequences: #1: 5′-GCACTCTTCATAACTCACTTT-3′; #2: 5′-GCACTGCTCATCAAAGATGTT-3′)(CloneID:TRCN0000092553; TRCN0000092556; GE Dharmacon, Lafayette, CO) with packaging vectors into 293T cells using FuGENE HD (E2311, Promega). Supernatants were concentrated using Fast-Trap Lentivirus Purification and Concentration Kit (FTLV00003, Millipore). Huwe1 knockdown in myoblasts were performed in the presence of 8 μg/ml polybrene at around 70% confluency followed puromycin (A11138-03, Life technologies) selection (1 μg/ml). Extensive passages of surviving cells were avoided in order to maintain knockdown efficiency. Huwe1 knockdown in myotubes was performed by incubating lentiviral concentrates in the presence of 8 μg/ml polybrene on differentiation day 4 for two days, followed by puromycin selection for another 2 days (1 μg/ml) to eliminate uninfected myotubes.

## Analysis of TBP protein half-lives using tandem fluorescent protein timers

The plasmid expressing wild type TBP-mCherry-sfGFP fusion protein under a CMV promoter were transfected into 293T cells by Fugene HD reagent following manufacturer's instructions. Cell imaging was done using Zeiss Axio Observer microscope. sfGFP signal was detected using excitation filter TBP 495/20, and mCherry signal was detected using excitation filter TBP 570/30. Same exposure time was used for different samples. Images were then further analyzed using ImageJ through the same processes. FACS analysis of TBP-mCherry-sfGFP signal was done in DB FACSAria III Cell Sorter. sfGFP was detected using 488 nm laser, and mCherry was detected using 561 nm laser. mCherry/sfGFP was set as a parameter and detected simultaneously with mCherry and sfGFP signals.

## Gene expression analysis

Total RNA from wild type and Huwe1 knockdown myoblasts or myotubes was isolated using RNeasy kit (74106, Qiagen, Valencia, CA). mRNA was then purified using Dynabeads Oligo(dT)$_{25}$ (25-61002, Life technologies). RNA-seq library was prepared using ScriptSeq v2 RNA-seq Library Preparation Kit (SSV21106, illumina, San Diego, CA), and then sequenced using an Illumina Hiseq 2000 sequencing platform. Reads were mapped to the genome using Tophat, and differentiation expression analysis was done using Cuffdiff (*Trapnell et al., 2009*, *2012*). Results were plotted out using R package Plotrix, and gene ontology analysis was done using R package Goseq (*Young et al., 2010*).

## Source files

The following data set was generated: Li L. A specific E3 ligase/deubiquitinase pair modulates TBP protein levels during muscle differentiation http://www.ncbi.nlm.nih.gov/geo/query/acc.cgi?acc=GSE72105. Publicly available at NCBI Gene Expression Omnibus.

## Acknowledgements

We thank Qing Zhong (UTSW) for pFastBac His-Huwe1, pCI-neo Huwe1 plasmid and sample Huwe1 protein for initiate testing; Lori Kohlstaedt and Vincent J CoatesProteomics/Mass Spectrometry Laboratory (P/MSL) for assistant with Mudpit analysis; Michael Knop (University of Heidelberg) for fluorescent timer plasmids and suggestions on fluorescent timer assays; Zhijuan Ma for flow cytometry experiments; Haiying Zhou for pCS2+ HA-TBP plasmid and suggestions on C2C12 experiments; Yick W Fong for suggestions and help for biochemical purification; Lingyan Jin for suggestions and help with in vivo ubiquitination assays; Claudia Cattoglio, Teppei Yamaguchi and Michael Rape for critical reading of the manuscript; Mallory Haggart, Shuang Zheng, Sarah Moorehead for assistance.

## Additional information

### Competing interests

RT: President of the Howard Hughes Medical Institute (2009-present), one of the three founding funders of eLife, and a member of eLife's Board of Directors. The authors declare that no competing interests exist.

### Funding

| Funder | Grant reference | Author |
| --- | --- | --- |
| Howard Hughes Medical Institute (HHMI) | | Robert Tjian |
| California Institute for Regenerative Medicine (CIRM) | | Robert Tjian |
| Howard Hughes Medical Institute (HHMI) | Fellow | Zhe Liu |

The funders had no role in study design, data collection and interpretation, or the decision to submit the work for publication.

### Author contributions

LL, Conception and design, Acquisition of data, Analysis and interpretation of data, Drafting or revising the article; SSM, WH, Acquisition of data, Contributed unpublished essential data or reagents; ZL, Conception and design, Drafting or revising the article, Contributed unpublished essential data or reagents; RT, Conception and design, Analysis and interpretation of data, Drafting or revising the article

### Author ORCIDs

Li Li, http://orcid.org/0000-0002-2981-6615

## Additional files

### Major dataset

The following dataset was generated:

| Author (s) | Year | Dataset title | Dataset ID and/or URL | Database, license, and accessibility information |
| --- | --- | --- | --- | --- |
| Li L | 2015 | A specific E3 ligase/ deubiquitinase pair modulates TBP protein levels during muscle differentiation | http://www.ncbi.nlm.nih.gov/ geo/query/acc.cgi? acc=GSE72105 | Publicly available at the NCBI Gene Expression Omnibus (Accession no: GSE72105). |

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
