## [Decision Letter]

Thank you for submitting your study titled “A Specific E3 Ligase/Deubiquitinase Pair Modulates TBP protein levels During Muscle Differentiation” for consideration in *eLife*. I am happy to report that your manuscript has been reviewed by three experts in the field and by a member of the Board of Reviewing Editors, who are excited about the findings reported in this manuscript. The reviewers have discussed the reviews with one another, and the Reviewing Editor has drafted this decision for your reference as you prepare a revised submission.

We look forward to receiving the revised paper addressing the following four comments from the three different reviews, which are appended below for your information.

1) Please measure the half-life of TBP via other methods (such as cycloheximide chase +/- RNAi against HUWE1 and USP10).

2) It will be very important to determine (if possible) the interaction interphase between TBP and Huwe1, and demonstrate that TBP that does not bind Huwe1, that it is not degraded during differentiation.

3) Please measure the dynamic changes of proteins involved in muscle differentiation (Pax7, *pax5*, MyoD, Myogenin) in control or Huwe1 targeted C2C12. This can be done at transcript levels or preferably at protein levels if antibodies are available.

4) In the Discussion section of the manuscript, please consider that the limits in the interpretation of the experiments need to be acknowledged. The analyses are limited to TBP and that the issue of TBP-associated factors from the other complexes remains open.

Reviewer #1:

In this manuscript, Tjian and colleagues identify the ubiquitin ligase Huwe1 and the ubiquitin protease USP10 as key factors responsible for the ubiquitin-proteasome system-mediated expression control of TBP during myogenesis. In addition, their results indicate that a fine-tune regulation of both enzymes is required for a correct muscle differentiation and myogenic gene expression program. Overall, this paper is clearly written and technically remarkable, in particular the biochemical part. However, a few points need to be clarified or modified before I can fully support the publication of this manuscript in *eLife*.

1) Figure 4 could be moved to a supplement and replaced by a cycloheximide chase that is more informative in term of protein half-life.

2) As mentioned in the text, TBP is not the unique target of Huwe1 and USP10 enzymes. Can an overexpression of TBP in C2C12 cells phenocopy Huwe1 knockdown or USP10 overexpression, at least on differentiation and maintenance of myotube morphology? In the absence of such a control, there is only a correlation between TBP expression level and myotube differentiation.

3) A USP activity, different from USP10 should be used as a negative control in Figure 7. A catalytic-dead mutant of USP10 (*cys* mutant) should also be used at least in Figure 7 and ideally in Figure 7.

Reviewer #2:

Li at el. described that during C2C12 muscle differentiation the protein level of TBP are high in proliferating myoblasts and declines during their differentiation to myotubes. Using HeLa cell extract and elegant biochemical purification they identified that Hewe1 is the ligase that regulates the level of TBP in this context. They also identified UbC5Hb/c as the E2s involved in Hewe1-mediated ubiquitylation. Interestingly, based on homology from yeast they identified USP10 as the iso-peptidase for TBP, and suggested that USP10 disassemble poly-ubiquitylated-TBP preventing the degradation of TBP. Thus, supporting but not directly testing the idea that a balance between the expression of Hewe1 and USP10 determines the level of TBP. Furthermore, loss of Huwe1 or to a lesser extent expression of USP10 resulted in a failure of C2C12 differentiation, and inability to express muscle specific genes as evident by RNA-seq and GO analysis.

The observations that the ubiquitin pathway directly regulates TBP level by a specific balance between DUB and ligase is novel, very important, with high relevance for the areas of transcription, protein dynamics, and as well as developmental biology.

However, the following major points need to be addressed experimentally prior to acceptance:

1) Determine the interaction interphase between TBP and Hewe1 as well as TBP and USP10 (as the author claims it directly interacts with unmodified substrate, Figure 7). Show that TBP that does not bind Huwe1, that it is not degraded during differentiation. In addition, the antagonisms between TBP and USAP10 as suggested by the authors may be very indirect. The ability of USP10 to degrade Huwe1-dependent poly-ubiquitylated TBP in vitro should be tested.

2) Huwe1, TBP, USP10 and the impact on transcription during differentiation:

A) The impact of Huwe1-dependent decline in TBP on transcription should be determined (e.g. a change from TATA to DPE/Inr mode of Pol-II recruitment). For example, can the authors test the impact of loss Hewe1 on TBP occupancy and Pol-II recruitment by ChIP-Seq during differentiation?

B) If indeed the authors claim for antagonistic function on TBP between USP10 and Huwe1 this should be reflected by antagonistic impact on a large body of genes when comparing RNA-Seq experiments (Figures 6 and 7). Simple bioinformatic analysis should address this point.

3) As correctly stated by the authors, the inhibition of muscle cell differentiation may be due to impact of Hewe1 on several other proteins than TBP, and that are required for muscle differentiation. Thus, this should not preclude an effort to established causality. An effort to directly connect the phenotypes associated with loss of Hewe1 to its regulation of TBP should be preformed. For example, (and in a simple model), it is possible to test if expression of a TBP mutant that is not recognized by the ligase (characterized above) would partially restore differentiation, or at least the expression of myogenic genes.

4) To support the expression analysis, and to better characterize Hewe1 impact on differentiation, the authors need to trace the dynamic changes at protein level of several muscle markers (Pax7, *pax5*, MyoD, Myogenin in control or Huwe1 targeted C2C12. Specifically in this experiment the level of MyoD are of interest, as it was shown that Hewe1 is involved in the degradation of MyoD in proliferating C2C12 myoblasts (Noy T et al., BBRC, 418 408-413). It will also be nice to show the protein level of Huwe1 and USP10 during C2C12 differentiation.

Reviewer #3:

The manuscript by Tjian and colleagues extend their earlier work on the switch in general transcription factor usage during myogenesis (9; 8). In this work they used the C2C12 cell system for myoblast to myotube differentiation to observe a switch from canonical 14-subunit TFIID to a heterodimeric TAF3-TRF3 complex. In differentiated myocytes TAF3-TRF3 takes over TFIID function to support the transcriptional program.

The present study investigates the molecular basis for this transcription factor switch. Using their C2C12 system, the authors employ a variety of approaches to conclude that the E3-HECT enzyme HUWE1 collaborates with Ubc5 enzymes to ubiquitylate TATA-binding protein (TBP) for subsequent proteosomal degradation. In addition, USP10 is identified as a TBP-specific deubiquitinase and counters TBP degradation and myocyte differentiation.

These conclusions are based on the following results:

a) cytoplasmic extracts from C2C12 cells prior and after terminal differentiation and from HeLa cells contain ubiquitination activity towards recombinant TBP;

b) highly-purified HeLa S100 fraction active in TBP ubiquitination contains the HECT-domain protein HUWE1;

c) recombinant HUWE1 can act as an E3 enzyme in TBP ubiquitination assays;

d) E2 enzymes of the UbcH5 family and UbcH7 can support HUWE1-dependent K48-linked ubiquitin on TBP;

e) HUWE1 knockdown increases TBP expression in 293T and C2C12 cells;

f) HUWE1 knockdown inhibits C2C12 differentiation at the cellular and gene expression level;

g) the deubiquitinase USP10 decreases in expression upon C2C12 expression;

h) overexpression of USP10 in 293T or C2C12 results in a small increase in TBP levels;

i) overexpressed TBP and USP10 can be co-immunoprecipitated in 293T cells.

There are several issues crucial to the conclusion that HUWE1/UbcH5 and USP10 control the switch from TBP and TFIID to TAF3-TRF3 that have not been resolved by the current study. As it stands now, the presented study is rather preliminary, for example:

1) It remains unclear whether HUWE1 and USP10 are the unique E3 and DUB enzymes in C2C12 cells directly controlling TFIID/TBP degradation. As stated by the authors, HUWE1 was one of the three E3s identified in the chromatographic fraction from HeLa cells. What were the others? Relevant in this is that the pattern of ubiquitylated TBP species is quite different when comparing reactions with the starting C2C12 extract (Figure 2) versus purified recombinant Huwe1 (Figure 3). It is difficult to ascertain whether fractions 35 and 36 are really peaking in the number of HUWE1 peptides. How many HUWE1 peptides are contained in side fractions 33 and 34? Did the authors go back and test the fractions mono Q fractions by HUWE1 immunoblotting? It is also important to note that recombinant HUWE1 seems much more processive (Figure 3) than observed in reactions with HeLa or C2C12 extracts.

An additional point of concern is that in order to monitor purification of the E3 from HeLa cells extracts the authors only used GST-TBP and TBP antibodies as detection. The authors should provide results form an in vitro ubiquitinylation experiment using HeLa fractions and the GST-TBP substrate probed with anti-Ubiquitin.

All these points raise significant doubt on the conclusion that HUWE1 is the relevant E3 activity.

2) The argument that a domain outside the HUWE1 HECT domain is required for TBP specificity, because the HECT-domain from another E3 E6AP does not support TBP ubiquitination, does not make sense. To make map TBP-specificity the authors should perform a structure-function study on HUWE1. In light of this, did the authors investigate whether TBP can interact directly with HUWE1?

3) To really verify that they isolated the C2C12-relevant E3 a control experiment using a C2C12 extract antibody-depleted for HUWE1 and/or USP10 should be provided to show that HUWE1 and USP10 are indeed the relevant enzymes as suggested by the RNAi experiments.

4) To assess whether TBP (and the TFIID TAFs) are indeed destabilized via proteosomal degradation in differentiated myocytes, the authors should determine the protein half-lives, preferably in a pulse-labelling setup. This should be combined with an RNAi against HUWE1 and against USP10 mRNA.

5) Are the TFIID TAFs also subjected to HUWE1/USP10-mediated control during myogenesis?

6) Are the TAF3 and TRF3 proteins resistant to HUWE1-mediated ubiquitylation? Or are they better substrates for USP10 deubiquitination, which leads to their protection from proteosomal degradation?

7) Relevant details about the part of HUWE1 expressed as a recombinant protein and the biochemical purity of the recombinant HUWE1 preparation should be provided.

8) Figure 8 does not help to clarify the text and is rather speculative at this stage, as it remains unclear whether ubiquitination and deubiquitination occur in the cytoplasm. It should be removed.

---

## [Author Response]

*1) Please measure the half-life of TBP via other methods (such as cycloheximide chase +/- RNAi against HUWE1 and USP10)*.

We have now measured the half-life of TBP protein using cycloheximide chase experiments and this data is included in the new Figure 5—figure supplement 1 (Control vs Huwe1 knockdown C2C12 cells) and the new Figure 7—figure supplement 2 (Control vs USP10 overexpressed C2C12 cells), showing that both Huwe1 knockdown and USP10 overexpression resulted in increased TBP protein half-life.

*2) It will be very important to determine (if possible) the interaction interphase between TBP and Huwe1, and demonstrate that TBP that does not bind Huwe1, that it is not degraded during differentiation*.

Following the reviewers’ suggestion, we have tried to characterize potential direct interactions between TBP and Huwe1. In the new Figure 3—figure supplement 2, we have now provided evidence that purified recombinant Huwe1 protein directly interacts with TBP through its DNA binding domain in vitro. Based on this result, it will be quite challenging to address whether TBP mutants that fail to bind Huwe1 are not degraded during differentiation. Previous crystallographic studies have shown that the conserved C-terminal DNA binding domain is a highly structured region of TBP (Nikolov et al., 1996). TBP mutants with truncated C-termini are most likely misfolded and will be degraded by Huwe1-independent pathways (Goldberg, 2003). Indeed, previous studies in yeast have shown that even point mutants in the TBP DNA binding domain can alter its protein stability (Jackson-Fisher et al., 1999; Kou et al., 2003). In any event, we have now acknowledged this limitation of our study in the Discussion section.

*3) Please measure the dynamic changes of proteins involved in muscle differentiation (Pax7,* pax5*, MyoD, Myogenin) in control or Huwe1 targeted C2C12. This can be done at transcript levels or preferably at protein levels if antibodies are available*.

We have now measured *Myod1*, *Myog* and *Mef2c* mRNA expression levels in control and Huwe1 knockdown C2C12 cells on differentiation Day0, Day3 and Day6 by RT-qPCR and included the data in the new Figure 6—figure supplement 2. Consistent with our phenotypic analysis and RNA-seq data, we observed inhibition of *Myod1*, *Myog* and *Mef2c* mRNA induction in Huwe1 knockdown cells during differentiation.

*4) In the Discussion section of the manuscript, please consider that the limits in the interpretation of the experiments need to be acknowledged. The analyses are limited to TBP and that the issue of TBP-associated factors from the other complexes remains open*.

We have now specifically pointed out that the regulation of TBP-associated factors remains unclear in the Discussion section.

References:

Goldberg, A.L. (2003). Protein degradation and protection against misfolded or

damaged proteins. Nature 426, 895-899.

Jackson-Fisher, A.J., Burma, S., Portnoy, M., Schneeweis, L.A., Coleman, R.A., Mitra, M., Chitikila, C., and Pugh, B.F. (1999). Dimer dissociation and thermosensitivity kinetics of the Saccharomyces cerevisiae and human TATA binding proteins. Biochemistry 38, 11340-11348.

Kou, H., Irvin, J.D., Huisinga, K.L., Mitra, M., and Pugh, B.F. (2003). Structural and

functional analysis of mutations along the crystallographic dimer interface of the

yeast TATA binding protein. Molecular and cellular biology 23, 3186-3201.

Nikolov, D.B., Chen, H., Halay, E.D., Hoffman, A., Roeder, R.G., and Burley, S.K. (1996). Crystal structure of a human TATA box-binding protein/TATA element complex. Proceedings of the National Academy of Sciences of the United States of America 93, 4862-4867.